# Green Nanoformulations of Polyvinylpyrrolidone-Capped Metal Nanoparticles: A Study at the Hybrid Interface with Biomimetic Cell Membranes and In Vitro Cell Models

**DOI:** 10.3390/nano13101624

**Published:** 2023-05-12

**Authors:** Alice Foti, Luana Calì, Salvatore Petralia, Cristina Satriano

**Affiliations:** 1Nano Hybrid Biointerfaces Laboratory (NHBIL), Department of Chemical Sciences, University of Catania, Viale Andrea Doria 6, 95125 Catania, Italy; alice.foti@phd.unict.it (A.F.); luanacali02@gmail.com (L.C.); 2Department of Drug and Health Sciences, University of Catania, Viale Andrea Doria 6, 95125 Catania, Italy; salvatore.petralia@unict.it

**Keywords:** plasmonic nanoparticles, gold nanoparticles, silver nanoparticles, palladium nanoparticles, supported lipid bilayers, prostate cancer cells, FRAP, FRET

## Abstract

Noble metal nanoparticles (NP) with intrinsic antiangiogenic, antibacterial, and anti-inflammatory properties have great potential as potent chemotherapeutics, due to their unique features, including plasmonic properties for application in photothermal therapy, and their capability to slow down the migration/invasion speed of cancer cells and then suppress metastasis. In this work, gold (Au), silver (Ag), and palladium (Pd) NP were synthesized by a green redox chemistry method with the reduction of the metal salt precursor with glucose in the presence of polyvinylpyrrolidone (PVP) as stabilizing and capping agent. The physicochemical properties of the PVP-capped NP were investigated by UV-visible (UV-vis) and attenuated total reflection Fourier transform infrared (ATR-FTIR) spectroscopies, dynamic light scattering (DLS), and atomic force microscopy (AFM), to scrutinize the optical features and the interface between the metal surface and the capping polymer, the hydrodynamic size, and the morphology, respectively. Biophysical studies with model cell membranes were carried out by using laser scanning confocal microscopy (LSM) with fluorescence recovery after photobleaching (FRAP) and fluorescence resonance energy transfer (FRET) techniques. To this purpose, artificial cell membranes of supported lipid bilayers (SLBs) made with 1-palmitoyl-2-oleoyl-sn-glycerol-3-phosphocholine (POPC) dye-labeled with 7-nitro-2-1,3-benzoxadiazol-4-yl (NBD, FRET donor) and/or lissamine rhodamine B sulfonyl (Rh, FRET acceptor) were prepared. Proof-of-work in vitro cellular experiments were carried out with prostate cancer cells (PC-3 line) in terms of cytotoxicity, cell migration (wound scratch assay), NP cellular uptake, and cytoskeleton actin perturbation.

## 1. Introduction

Cancer is the main cause of morbidity and mortality worldwide, excluding infectious diseases, and is one of the most challenging diseases to treat, because cancer is characterized by the development of mutated cells that divide frenziedly, thenceforth they can spread to different organs in a process called metastasis [1]. Hence, the application of nanomaterials and nanotechnologies to target cell migration for the development of new anti-inflammatory and antitumor-based drugs is particularly promising.

Noble metal nanoparticles (NP), owing to the characteristic surface plasmon resonance (SPR), as a result of their interaction with light, which results in the different colors of the colloidal solutions depending on the particle size and shape, together with properties of high surface area, multi-functionalization, highly stable nature, and easy processing, have led to unprecedented exploitation of nanomaterial research applications in biomedicine and cancer treatments, and their use as innovative chemotherapeutic drugs are on the rise in cancer research [2].

Among all metal nanoparticles, gold (Au), silver (Ag), and palladium (Pd) NP have gained specific attention for nanomedicine and theranostic applications in the treatment of cancer [3,4].

AuNP of various shapes are of particular interest for the application in photothermal therapy (PTT), due to their distinctive properties, such as absorption and scattering of electromagnetic radiation with a wide spectrum of absorption maxima (from ultraviolet to the near IR) and high absorption cross sections reachable simply by size, shape, and chemical composition tuning [5]. Depending on the nanoparticle size and surface charge, AuNP exhibits anti-angiogenic properties, e.g., by inhibiting the function of pro-angiogenic heparin-binding growth factors by triggering their denaturation [6], oxidative stress, membrane, and DNA damage [7]. Due to the leaky lymphatic flow and angiogenic vessels in cancerous cells, AuNP, as many other nanoparticles, can accumulate more or selectively inside cancerous cells as compared to the normal cells [8], which is a process of passive targeting known as enhanced permeability and retention (EPR) effect in cancer treatment. AuNP was demonstrated to inhibit tumor growth and metastasis in models of ovarian cancer in a size- and concentration-dependent manner, by down-regulating MAPK-signaling, and reversing the epithelial–mesenchymal transition by reducing the secretion of many proteins involved in the process [9]. In addition, AuNP can impair cancer cells through the inhibition of cell metastasis and migration [10,11,12]. For example, Palaniyandi et al., showed that spherical AuNP with an average size of 127 nm exhibits an anti-metastatic activity against human fibrosarcoma cancer cell line by disturbing the actin polymerization pathway, which in turn inhibits the chemo-attractant cell migration [13]. In a previous work, we investigated the cytotoxicity and the perturbation on cytoskeleton actin of hyaluronan (HA)-functionalized spherical AuNP, about 10 nm in size, on CD44-positive prostate tumor cells in comparison with neuroblastoma cells, which do not express the CD-44 HA binding receptor; a receptor-dependent perturbation effect by HA-AuNP on cytoskeleton actin and lysosomal organelles was demonstrated [14].

AgNP has intrinsic antibacterial and anticancer properties, which can be further enhanced by the nanoparticle functionalization with functional biomolecules [15]. Following the uptake by endocytosis-mediated mechanisms [16], AgNP are directed from endosomes, where they originally gather, to lysosomes, where the acidic environment leads to an increased release of Ag+ from the nanoparticle. The reactive Ag+ ions then unbalance cellular homeostasis and can lead to the cellular apoptosis [17]. Other mechanisms of cytotoxicity include the displacement of Ag^+^ ions or the AgNP as a whole, where the cellular uptake of AgNP, which lead to the deregulation of important cellular mechanisms, and are linked to DNA damage, change in the intake of nutrients, the stimulation of intracellular reactive oxygen species (ROS)-associated membrane damage, and cell-cycle arrest, depend on properties such as size, dose, route of administration, and the capping agents [18]. Among mechanisms of NP-induced cell death being studied, apoptosis and autophagy, and their cross-talk, are widely accepted signaling pathways for the death of malignant cells as an outcome of triggers initiated by internal or external stimuli [19,20]. AgNP can act as an anti-angiogenic factor by targeting the activation of PI3K/Akt signaling pathways [21]. The antimetastatic potential of AgNP has been observed against colorectal and breast cancer models, with the cell migration inhibition in a concentration-dependent manner and a 150–250% rise in the production of ROS [22].

PdNP have shown a significant potential for the development of nanomedicine with characteristic antimicrobial, wound healing, antioxidant, and anticancer properties. Biogenic PdNP are biocompatible, widely distributed, and generally preserve the anticancer properties of the plant extracts and microorganisms such as bacteria, algae, fungi, and viruses used for the synthesis [23]. Moreover, PdNP’s intrinsic anticancer properties include DNA and protein damage and inactivation [24,25], the leakage of lactate dehydrogenase [26], and the production of free radicals of both ROS and reactive nitrogen species (RNS) that cause lipid degradation [24]. The 24 h treatment of ovarian cancer A2780 cells with PdNP of 5 nm in size revealed dose-dependent toxicity of the PdNP, increased ROS generation, NP-induced autophagy, impairment of mitochondrial membrane potential, and enhanced caspase-3 activity [26].

The use of capping agents is very frequent in the synthesis of colloidal dispersions to precisely control the NP growth, aggregation, and physicochemical characteristics. The surface capping of nanoparticles is very relevant to enhance their biological properties. For instance, the use of polyvinylpyrrolidone (PVP) as capping agent for metal nanoparticles such as Fe, Ag, Au, Zn, etc., offer many advantages to make them attractive candidates for biomedical applications such as drug delivery and theranostics in cancer [27]. PVP-functionalized PdNP were demonstrated to exhibit excellent cytotoxic activity against human breast cancer cells in a dose-dependent manner, by generating ROS in the mitochondria, which leads to the damage of the mitochondrial membrane potential and nuclear DNA, and induces apoptosis through caspase3/7 enzymatic activity [28]. We recently demonstrated the anti-tumor and anti-metastatic potential in prostate cancer (PC-3) cells of a nano-delivery system based on PVP-capped PdNP and cisplatin [25].

This study was aimed at the design, synthesis, and characterization of PVP-capped gold, silver, and palladium NP using a green and environmentally friendly approach, to test their effect on the lateral diffusion properties of artificial cell membranes made of supported lipid bilayers (SLB), as well as in terms of cell migration and cytoskeleton actin in prostate cancer (PC-3) cells. To the best of our knowledge, there are no similar results reported earlier in the literature.

## 2. Materials and Methods

### 2.1. Chemicals

Gold (III) chloride trihydrate (HAuClO_4_·3H_2_O, purity ≥ 99.9%), silver nitrate (AgNO_3_), palladium chloride anhydrous (PdCl_2_), sodium hydroxide (NaOH, purity ≥ 97%), D-(+)-glucose (purity ≥ 99.5%), polyvinylpyrrolidone (PVP, average molecular weight: 44,000 Da) were purchased from Sigma-Aldrich (St. Louis, MO, USA). Hydrochloric acid (HCl, 37% in H_2_O) was purchased from (Carlo Erba, Milan, Italy). The glassware was cleaned with aqua regia (HCl: HNO_3_ 3:1 volume ratio) and rinsed with ultrapure Milli-Q water before each use. For all the experiments, ultrapure Milli-Q water was used (18.2 mΩ·cm at 25 °C, Millipore, Burlington, MA, USA); 1-palmitoyl-2-oleoyl-sn-glycerol-3-phosphocholine (POPC), 1,2-dioleoyl-sn-glycero-3-phosphoethanolamine-N-(7-nitro-2-1,3-benzoxadiazol-4-yl) (ammonium salt) (NBD PE), and 1,2-dioleoyl-sn-glycero-3-phosphoethanolamine-N-(lissamine rhodamine B sulfonyl) (ammonium salt) (Rh PE) were purchased from Avanti Polar Lipids (Birmingham, AL, USA).

RPMI-1640 medium, Dulbecco’s Modified Eagle Medium (DMEM)-F12, penicillin-streptomycin, L-glutamine, fetal bovine serum (FBS), dimethyl sulfoxide (DMSO), Dulbecco’s phosphate-buffered saline (PBS), Triton X-100, bovine serum albumin (BSA), and paraformaldehyde were purchased by Sigma-Aldrich (St. Louis, MO, USA). Hoechst 33342 Solution, MitoTracker™ Deep Red FM, ActinGreen 488 ReadyProbes Reagent, and paraformaldehyde were purchased by ThermoFisher Scientific (Waltham, MA, USA).

### 2.2. Synthesis of PVP-Capped Metal Nanoparticles

All metal nanoparticles were synthesized by a green reduction method based on glucose and PVP as reducing and capping/stabilizing agents, respectively.

AuNP and AgNP were synthesized by modifying the procedure reported by Sanfilippo et al. [14]. Briefly, 76 μL of 1 mM PVP was added to 1.434 mL MilliQ-H_2_O. Then, 200 μL of 10 mM HAuCl_4_ (or AgNO_3_) was added to the solution and stirred for 10 min at room temperature. Afterward, 100 μL of 100 mM D-(+)-glucose was added and the solution was left under stirring for about 15 min. Finally, 210 μL of 100 mM NaOH was added dropwise to promote Au(III) reduction to Au(0) (or Ag(I) to Ag(0)) with the oxidation of glucose to corresponding gluconic acid. The NP were formed after a few minutes, as observed by the eye as the colorless chloroauric acid (or silver nitrate) solution altered to red for Au NP (or yellowish-brown for AgNP). The obtained colloidal dispersion was purified by removing the reactants’ excess through two centrifugation steps (10,000× rpm, 30 min, 29 °C), with water washing in between. The final sample, named ‘pellet 2’, was collected and stored in the dark at 25 °C.

PdNP were synthesized by modifying the procedure reported by Xu et al. [29]. Briefly, 60 mM palladium salt solution was prepared by dissolving 1.7 mg of PdCl_2_ in 165 µL of 0.2 M HCl into a 4 mL glass vial at 60 °C, under stirring, until the metal salt was completely dissolved, as visible by the formation of a light orange-colored clear solution. A 76 μL of 1 mM PVP was added to the solution as the stabilizing reagent. The prepared mixture was left to cool to room temperature. Afterward, to reduce Pd(II) to Pd(0), a volume of 250 µL of D-(+)-glucose 2 M aqueous solution was added under moderate stirring. Finally, 1.0 M NaOH was added dropwise to the resulting dark orange-colored solution until reaching neutral pH. To obtain the purified ‘pellet 2’ sample, the as-prepared PdNP were centrifuged twice using Amicon tubes (Merk Life Science S.r.l., Milan, Italy) with a 30 kDa MW cut-off filter (8000× rpm, 3 min, 29 °C), with ultrapure Milli-Q water washing in between.

### 2.3. Preparation of Small Unilamellar Vesicles (SUVs) and Supported Lipid Bilayers (SLBs)

Lipid vesicles of Rh PE (and/or NBD PE)-labeled POPC were synthesized using POPC and Rh PE (and/or NBD PE) lipid chloroform solutions as previously described [30]. Briefly, POPC was mixed with Rh PE (and/or NBD PE) in a round-bottomed flask to obtain, in a volume of 1 mL of chloroform, phospholipid vesicles at the final concentration of 5 mg/mL, with 2% wt. of dye. Due to the light sensitivity of the dye-labeled system, the flask was maintained covered under an aluminum foil for the whole preparation procedure, from the initial steps to the final storage. The chloroform was evaporated inside a cabinet under N_2_ flux, while rotating the round-bottomed flask, to obtain a uniformly deposited dry lipidic film on the wall of the flask. Residual chloroform was removed by leaving the flask for 30 min in a rotating evaporator at room temperature. The lipidic film was emulsified in filtered and degassed buffer (PBS, 10 mM, pH = 7.4), vortexed and extruded 11 times through a 100 nm polycarbonate membrane, followed by another 11 times through a 30 nm membrane (Avanti Polar Lipids Inc., Alabaster, AL, USA), to narrow vesicle size distribution and obtain SUVs with a diameter of about 100 nm. The final product was stored in the dark at 5 °C, in a glass vial insufflated with N_2,_ and sealed with parafilm.

To fabricate SLBs by the process of SUVs’ absorption–rupture–fusion [31], glass surfaces (22 mm glass bottom dishes, Willco Well, Amsterdam, The Netherlands) were hydrophilized by UV/ozone treatment (2 cycles of 15′ each, with water rinsing and N_2_ blow-drying of the glass substrates before and in between). Immediately after the UV/ozone treatment, the glass substrates were washed with PBS and incubated 20′ with the SUV sample at the final concentration of 100 µg/mL.

### 2.4. Cellular Maintenance and Treatments

A prostate cancer PC-3 cell line was cultured in RPMI-1640, supplemented with 10% *v/v* fetal bovine serum (FBS), 2 mM L-glutamine, 50 IU/mL penicillin, and 50 μg/mL streptomycin. Cells were grown in tissue culture-treated Corning^®^ flasks (Sigma-Aldrich, St. Louis, MO, USA) in an incubator (Heraeus Hera Cell 150C incubator, Heraeus S.p.A., Cavenago di Brianza (MB), Italy), under a humidified atmosphere at 37 °C in 5% CO_2_.

### 2.5. Physicochemical and Biophysical Characterization, In Vitro Cellular Analyses

#### 2.5.1. UV-Visible (UV-Vis) Spectroscopy

UV-vis spectroscopy was performed on a Perkin Elmer UV-vis spectrometer (PerkinElmer 365, PerkinElmer, Waltham, MA, USA), in the wavelength range of 200–700 nm. UV-vis spectra of the aqueous dispersions of all metal nanoparticles were recorded using quartz cuvettes with an optical path length of 0.1 cm.

#### 2.5.2. Attenuated Total Reflectance Fourier Transform Infrared (ATR-FTIR) Spectroscopy

ATR-FTIR spectroscopy (PerkinElmer Frontier FT-IR Spectrometer with Spotlight 400 and ATR accessory) was performed at room temperature in the wavenumber range between 450 to 4000 cm^–1^. The spectra were collected as a result of 32 running scans at a resolution of 4 cm^−1^.

#### 2.5.3. Dynamic Light Scattering (DLS) and Zeta Potential (ZP) Analyses

DLS and ZP measurements were carried out with a ZetaSizer NanoZS90 Malvern Instrument (Malvern, UK), equipped with a 633 nm laser (scattering angle = 90°, T = 25 °C) to evaluate hydrodynamic size distribution and surface charge of the nanoparticles, respectively.

Nano Tracker Analysis was used to record the Brownian motion of the colloidal particles and to analyze their size distribution. The measurements were repeated three times as soon as the injection of 1 mL of the sample into the cell equilibration was reached. The results are presented as the mean ± standard deviation (SD) of at least three measurements.

#### 2.5.4. Atomic Force Microscopy (AFM)

AFM images were acquired using a Cypher AFM instrument (Asylum Research, Oxford Instruments, Santa Barbara, CA, USA) equipped with an XY scanner with a scan range of 30/40 μm (closed/open loop) and operating in AC-mode imaging in air. Aluminum reflex one-side coated silicon rectangular 30 μm long cantilevers, with tetrahedral tips, were purchased from Olympus (AC240TS, Oxford Instruments, Abingdon, UK). The probes had nominal driving frequency and spring constant values of 70 kHz and 2 N/m, respectively. Samples were prepared by dropping 10 μL of an aqueous dispersion of the NP on freshly-cleaved muscovite mica (Ted Pella, Inc., Redding, CA, USA). After 5 min of incubation, samples were washed with ultrapure Milli-Q water and dried under gentle N_2_ flow. Images were acquired at scale sizes of 2 and 1 μm^2^ and then analyzed by using a free tool in the Asylum Research offline section analysis software (version 16).

#### 2.5.5. Confocal Laser Scanning Microscope (LSM)

A confocal microscope (FV1000 LSM system Olympus, Shinjuku, Japan), equipped with spectral filtering and four lasers (diode UV: 405 nm; multiline Argon: 457, 488, and 515 nm; HeNe green: 543 nm; HeNe red: 633 nm) was used with an oil immersion objective (60× O PLAPO). A fixed value was defined for the detector gain, and images were collected, in sequential mode, randomly all through the area of the sample.

#### 2.5.6. FRAP and FRET Analyses

After the incubation with SUVs and the SLB formation, the samples were rinsed with buffer and imaged with the LSM both before and after the addition, directly in the Petri dish on the microscope stage, of the Au, Ag, and PdNP pellet 2 samples (20× diluted in PBS). For confocal FRAP analysis, time-solved snapshots of 256 × 256-pixel scans were collected at an interval of 5′′ each from the other, both before and after the bleach (intense illumination with Ar laser at 95% of power) in a region of interest (ROI) region with a radius of 10.35 µm. After the photobleaching, the monitoring of fluorescence intensity recovery in the bleached ROI as a function of time allow to calculate the diffusion coefficient (D), as quantitatively stated in the Axelrod’s equation modified for confocal FRAP data obtained using circular bleach regions [32]:(1)$$D=0.25\frac{r_{n}^{2}}{\tau_{1/2}},$$
where $\tau_{1/2}$ is the half-time of recovery, defined as the time required for a bleach spot to recover halfway between initial and steady-state fluorescence intensities, and $r_{n}$ is the nominal radius of the bleach spot. Another parameter that can be calculated from FRAP analysis is the ratio ($M_{f}$) between the mobile and immobile fractions of molecules, defined as:(2)$$M_{f}=\frac{I_{\infty }-I_{0}}{I_{t=0}-I_{0}}$$
where *I*_∞_ is the final recovered intensity, *I*_0_ is the intensity value immediately after bleach, and *I_t_*_=0_ is the pre-bleach intensity value. 

The FRAP image processing and analysis were performed with ImageJ software, by utilizing the FRAP profiler macro, with data normalized to the initial (pre-photobleach) value. For each sample, the emission recorded from the bleached ROI was compared with that coming from contiguous non-bleached areas.

For confocal FRET imaging, the filters were removed, an apt average power was used to reduce photobleaching, and, to not saturate the pixel intensity, the gain of the photomultiplier tube (PMT) was fixed alike for both donor (*D*) and acceptor (*A*) emission channels. The following three classes of images were acquired: (1) *D*- and *A*-channel images from the single-labeled donor lipids excited with donor molecule excitation wavelength; (2) the A-channel images acquired from the single-labeled acceptor molecule excited with donor and acceptor wavelength; (3) the *D*- and *A*-channel images acquired from the double-labeled (*D* + *A*) lipids excited with donor excitation and with acceptor excitation wavelengths. These images were used to obtain the processed FRET images with the Olympus FV1000 software (version 4.2b) and, quantitatively, the efficiency of energy transfer *E*, was evaluated as:(3)$$E=1-\frac{t_{DA}}{t_{D}},$$
where $t_{DA}$ is the donor lifetime in the presence of the acceptor and $t_{D}$ is the donor’s lifetime in the absence of the acceptor.

The Förster length ($R_{0}$), which refers to the separation distance of a single FRET pair corresponding to 50% energy transfer efficiency, the value of 6.6 nm was used for the NBD-PE/Rhod-PE FRET pair [33].

#### 2.5.7. MTT Assay

PC-3 cells were seeded at a density of 1.0 × 10^4^ cells/well in a 96-well/plate and maintained in their respective complete media in standard culture condition. The day after, cells were washed with 1% FBS-supplemented medium and treated with samples with a 20×, 50×, and 100× dilution of pellet samples. After 24 h of incubation, cells were treated with 5 mg/mL of 3-(4,5-dimethyl-2-thiazolyl)-2,5-diphenyl-2H-tetrazolium bromide (Sigma-Aldrich, St. Louis, MO, USA) at 37 °C for 180 min. At this stage, the formazan salts formed by succinate dehydrogenase activity in live cells were solubilized with DMSO and quantified spectrophotometrically by a Synergy 2 microplate reader (BioTek, Winooski, VT, USA), by the absorbance value at 570 nm of wavelength. All conditions were measured in triplicate and results were expressed as % of viable cells over the negative control (i.e., untreated cells).

#### 2.5.8. Confocal Microscopy Imaging of Cellular Uptake and Cytoskeleton Actin

PC-3 cells were seeded at a density of 2.0 × 10^4^ cells per dish in 12 mm diameter glass bottom dishes (WillCo-dish^®^, Willco Wells, B.V., Amsterdam, The Netherlands) in 10% *v*/*v* FBS-supplemented RPMI-1640 complete medium.

After 48 h, cells were rinsed and, subsequently, incubated for 2 h with AuNP, AgNP, and PdNP pellets (pellet 2 sample, 20× dilution) in the complete medium supplemented with 1% *v*/*v* FBS. At the end of the incubation time, the cells were washed with 10 mM PBS (37 °C, pH = 7.4) and stained for 15′ with 1 μg/mL of Hoechst 33, 342, and 200 nM MitoTracker Deep Red. Afterward, cells were washed with PBS and fixed with high-purity paraformaldehyde (4% *w*/*v* in PBS) for 15′. Finally, cells were rinsed with PBS and left in PBS.

For cytoskeleton actin staining, the fixed PC-3 cells were permeabilized by using 0.02% *w/v* Triton X-100 and 10% bovine serum albumin (BSA). Then, they were treated for 30′ with a high-affinity F-actin probe, conjugated to green-fluorescent Alexa Fluor^®^ 488 dye (ActinGreen™ 488 ReadyProbes^®^ Reagent, ThermoFisher). In the end, cells were rinsed with PBS and left in PBS. Confocal images were recorded in random fields of the samples prepared in triplicate. The LSM micrographs were deconvoluted and quantitatively analyzed in terms of the sum of all voxel values with Huygens Essential software X11 version 22.10 (by Scientific Volume Imaging B.V., Hilversum, The Netherlands). A one-way ANOVA test was used for the related statistical analysis.

#### 2.5.9. Cell Migration (Wound Scratch Assay)

PC-3 cells were seeded at a density of 1.0 × 10^6^ cells/well in 6-multiwell plates, and cultured in their complete medium until confluence. At that point, the cell monolayers of were scratched and wounded using a sterile universal 10 μL pipette tip. Immediately after the scratch, cells were rinsed with the complete medium, and plates were marked to ensure that scratches were measured at the same location for each experiment. Next, every well was treated with the NP (pellet 2 samples, 50× dilution) in 1% *v*/*v* FBS-supplemented RPMI 1640 medium. A parallel experiment with 760 nM PVP treatment (positive control) was carried out.

Serial phase-contrast images (Leica, Wetzlar, Germany) of the in vitro wounds were taken immediately after the treatment, and after 3, 6, 24, and 48 h of incubation, the separation wall width was measured by using the MRI Wound Healing Tool on the ImageJ software (version 1.50i, NIH).

## 3. Results and Discussion

### 3.1. Physicochemical Characterization of PVP-Capped Metal Nanoparticles

The UV-visible spectra of AuNP, AgNP, and PdNP are shown in Figure 1. In Figure 1a, reporting the plasmonic bands for as-prepared and purified (pellet 2 sample) nanoparticles, one can see the plasmon bands centered approximately at 262 nm for PdNP, 397 nm for AgNP, and 517 nm for AuNP. According to the literature, the formation of spherical nanoparticles can be assumed, with an optical size (d_o_) approximately 2 nm [29], 14 nm [34], and 5.4 nm [35], respectively. The spectra of the purified NP (dashed lines) do not show any significant shift in the plasmon wavelength with respect to those of as-prepared NP (solid lines); this finding indicates stability in the size after the centrifugation steps.

To note, the centrifugation steps used for the purification were also effective to concentrate the NP in the respective pellets, as indicated by the hyperchromic shift of the plasmon bands of about three times for both PdNP and AgNP and more than eight times in the case of AuNP. Table 1 provides the quantitative analysis of as-prepared and pellet 2 fresh samples in terms of optical diameter and extinction coefficient.

In the SPR band, agglomeration of NP should result in the plasmon resonance wavelength shift to longer wavelengths (‘red-shift’), because placing metal NP close to each other can have an outcome in the coupling of their plasmons [36]. Figure 1b–d show that all the NP resulted in being general and very stable upon aging up to four weeks (see timepoints from 0 to 3 indicated in the figure). However, AuNP spectra (Figure 1b) exhibited no significant wavelength shifts but only a slight hypochromic shift; whereas in the case of AgNP (Figure 1c) and PdNP (Figure 1d), red shifts up to, respectively, ~5 nm and ~10 nm were detected, which indicate some degree of aggregation. Such a finding can be explained by the presence of Ag^+^ and Pd^2+^ ions, released, leached, or dissolved depending on the oxidation of metallic nanosilver [37] and nano palladium [25] by dissolved oxygen and protons, while nanogold is much more stable.

In Figure 2 are reported the FTIR spectra of the purified NP, in comparison with the reference spectrum of the PVP alone. The PVP-characteristic peaks at 1644 cm^−1^ and 1289 cm^−1^, ascribed respectively to the C=O and –C-N bond stretching vibrations, are shifted to 1648 cm^−1^ and 1293 cm^−1^ in the PVP-capped Ag and Pd NP. Such a trend is explained by the coordination of Ag^+^ ions and Ag atoms to carbonyl oxygen or nitrogen atoms arising from PVP units [41,42] and the chelating-chemisorption process of PVP molecules onto the surface of Pd [43]. No significant peak shifts were instead detected for PVP-capped AuNP, although a shift in the C=O stretching, due to intermolecular hydrogen bonding [44], could be expected for a significant interaction at the metal–polymer interface. The peak at 1353 cm^−1^, assigned to the C−H bend vibrational modes of CH_2_ moieties in PVP is strongly decreased or disappeared in all the PVP-capped NP. Finally, for all three NP, and especially in the case of PdNP, the absorption of the glucose-specific region (1100–1000 cm^−1^) [29] is observed.

The AFM analysis of the purified NP confirms the formation of nanoparticles spherical in shape (Figure 3) with average size significantly different for the diverse metals. Indeed, the particle size analysis evaluated by the height value of the AFM images (Table 2) points out to the values of about 9 nm for AuNP, 5 nm for AgNP, and 2 nm for PdNP, respectively. Another interesting finding, especially evident from the amplitude images, is that both AuNP and AgNP but not PdNP display a visible shell around the metallic core (see insets in Figure 3).

The PVP chains contain C=O, C-N, and CH_2_ functional groups, a strongly hydrophilic component (the pyrrolidone moiety), and a hydrophobic group (the alkyl group). As a result of the highly polar amide group within the pyrrolidone ring, the PVP is an excellent solvent in water and can act as a stabilizer preventing the aggregation of NP by steric hindrance effect. The PVP–metal interaction is expected to occur through the carbonyl oxygen or nitrogen atoms of the repeating unit and the metal surface [45]. In some cases, the interparticle distances are so elongated that PVP can be considered the dispersant phase [45]. According to the AFM analysis, we can figure out that the polymer is acting as a capping agent for gold and silver NP and as a dispersant matrix for PdNP.

The DLS and ZP analyses reported in Table 2 suggest that the stability of the colloids can be ruled out in terms of different mechanisms.

In the case of the small PdNP, the ZP value of ~−41 mV, indicate a predominant electrostatic mechanism. Indeed, the dispersion of particles can typically be considered stable if the zeta potential results are more positive than +30 mV or more negative than −30 mV, as a result of inter-particle electrostatic repulsion [46]. For the larger AuNP and AgNP, in both cases, a zeta potential value close to zero suggests that the polymer chains coated the NP neutralizing the charge yet the samples are stable mainly due to PVP steric stabilization [47].

To note, the hydrodynamic sizes of 29 ± 10 nm for AuNP and 5 ± 2 nm for AgNP point to a thin and quite bounded polymer shell around these nanoparticles. On the other hand, the d_H_ value of 45 ± 1 nm indicates a thicker shell and an expanded hydrodynamic radius.

Indeed, the PdNP smaller particles offer higher surface curvatures than larger AuNP and AgNP, and therefore, are expected to allow an increased polymer loading per unit surface area. The polymer conformation gradually changes from collapsed to extended conformation as the surface density increases [48]. Consequently, when surface density increases, that is, the nanoparticle size decrease, we expect the PVP chains to elongate and orient with respect to the nanoparticle surface normal, resulting in a thicker and less hydrated polymer coating around the PdNP, which is consistent with the hypothesis of the PVP role of dispersant matrix more than the capping agent as discussed above.

### 3.2. Biophysical Studies of the Nano–Bio Interface between PVP-Capped Metal NP and SLBs

To scrutinize the perturbative effect of the three NP types onto model cell membranes, FRET and FRAP experiments were carried out on SLBs.

Figure 4 shows the results of FRET analyses for energy transfer processes investigated for SLBs labeled with NBD–Rh FRET donor-acceptor pair, before and after a short time of interaction (5–10′ since the addition) with the nanoparticles. While AgNP does not induce any statistically significant change in the efficiency of FRET transfer (ΔE, Figure 4a) nor in the average spatial distance (ΔR, Figure 4b) of the probes with respect to the membrane before these interactions (SLB), both AuNP and PdNP decrease FRET, with the average spatial separation of the probes being increased by ~7 Å or 6 Å for SLB + AuNP or SLB + PdNP, respectively.

The FRAP analysis of the NBD-Rh-SLBs, with calculated D values, respectively, of 1.31 ± 0.02 µm^2^/s for NBD and 1.23 ± 0.07 µm^2^/s for Rh dye probe-labeled membranes, did not show significant changes in the average diffusion coefficients after the interaction with the NP. However, the averaged curves of the intensity for three FRAP experiments with SLB, before and after the interaction with AuNP, AgNP, or PdNP (Figure 5) reveal some interesting features. The recovery curves with the lowest value for the mobile fraction M (accounting for the ability of the lipid molecules within the supported bilayer to diffuse during the duration of the experiment) is that of SLB + PdNP, followed by that of the bare SLB, while SLB + AuNP and SLB + AgNP samples exhibit comparable higher mobile fractions M.

Although it has no direct relation with biophysical parameters, this finding is remarkable, as a semi-quantitative estimate of molecule dynamics and can be used to compare the interaction with the three diverse nanoparticles. The ratio of mobile to the immobile fractions, M_f_ (see Equation (2)), for the SLBs treated with the three types of NP shows an increase in the estimated value for the membranes treated with PdNP but no evident changes for AuNP- nor AgNP-treated SLBs (Table 3).

### 3.3. Cellular Studies of the Interaction between PVP-Capped Metal NP and PC-3 Cells

One of the tumor malignancies that would benefit from nanomedicine-based strategies to counteract metastasis and angiogenesis is prostate cancer. Prostate cancer is known to have a high chance of developing metastasis, even at the early stages, hence new anti-metastatic potential therapies are needed [49,50]. While locally advanced prostate cancer has been successfully treated with surgical resection, patients diagnosed with metastatic disease must rely on therapeutic approaches often associated with systemic side effects [51], which, instead, nanomaterials can minimize [52].

The cell viability of the chosen cell line (PC-3) was investigated to assess the cytotoxicity of our nanoparticles. Figure 6 shows that only AuNP and Ag present significant toxicity at the highest concentration tested in the present study. On the other hand, PdNP show no toxicity.

The size, charge, shape, and bioconjugation of NP rule out their uptake via macropinocytosis, diffusion, or receptor-mediated mechanisms [15]. Recently, AuNP conjugated with a prostate-specific membrane antigen were demonstrated to be internalized by prostate cancer cells via receptor-mediated endocytosis [53]. The cellular uptake of AgNP occurs through endocytosis, then NP are encapsulated in lysosomes, where, due to the acidic environment and a variety of hydrolytic enzymes, they suffer degradation of capping molecules and release Ag^+^. In turn, this results into the increase of ROS, including superoxide anions (O_2_^−^), hydroxyl radicals (^•^OH), and hydrogen peroxide (H_2_O_2_), as well as the trigger of mitochondrial damage and inducing of apoptosis [54]. However, the toxicity of AgNP on a given cancer cell depends rather on the uptake efficiency of the intact nanoparticles than on the sensitivity of the cell to Ag ions [55]. AgNP with a glucose functionalization resulted effective in favoring the uptake of the AgNP by prostate cancer cells [56].

To evaluate the inhibitory effect on cancer cell migration of the three NP, the wound scratch assay was performed on the PC-3 prostate cancer line and micrographs were acquired with the optical microscope until wound closure was reached. The results are summarized in Figure 7. Noteworthy, the cells treated with the PVP alone (positive control) exhibit a significantly higher migration after both 3 and 6 h after the scratch than the control untreated ones, while this effect is not observed for the PVP-capped NP. Indeed, a significative reduction in cell migration was found for PdNP-treated cells, especially after 24 h. A less effective, but still significative, inhibition of cell migration was found also for AuNP. No significant differences with respect to the control untreated cells were observed for the treatment with AgNP.

Taking into account the more ‘exposed’ polymer chains in the case of PVP-capped PdNP (see discussion above), the results are not surprising, as recent findings demonstrated the prevention of tumor cell migration by PVP-based hydrogels [53]. Concerning AuNP, from the literature, it is also known that they can slow down the migration/invasion speed of cancer cells and suppress metastasis [54]. As an example, Mahalunkar et al., prepared folate–curcumin-loaded gold–PVP nanoparticles and observed a high anti-migratory potential in human breast epithelial and mouse fibroblast cell lines [55]. Furthermore, AuNP was demonstrated to suppress proliferation, migration, and invasion in papillary thyroid carcinoma cells via the downregulation of CCT3 [10]. Bellissima et al. found, especially after short treatment times, an inhibitory effect in cell migration on PC-3 treated with PVP-capped PdNP, in correlation with increases in ROS production and cytotoxicity tested by MTT [25].

Figure 8 shows the representative LSM micrographs of PC-3 cells untreated (Figure 8a,e,i) or 2 h treated with AuNP (Figure 8b,f,j), AgNP (Figure 8c,g,k), or PdNP (Figure 8d,h,l).

For all the NP-treated cells, the merged bright-field micrographs with the confocal images of Hoechst-stained cells (Figure 8a–d) show the perinuclear accumulation of NP, identified by the dark spots (yellow arrows). The corresponding mitochondrial staining by MitoTracker Deep Red (Figure 8e–h) evidenced a perturbation of the organelles, which are less defined in the NP-treated cells with respect to the untreated cells. As to the F-actin staining with Actin Green Probe, actin filaments and long stress fibers look thicker for the NP-treated cells (Figure 8j–l) than the thinner filaments (gray arrows) of control untreated cells (Figure 8i), with less evidence of protrusions, especially in the case of PdNP-treated cells.

Mitochondria are very central organelles that affect tumorigenesis and metastatic dissemination through different mechanisms including regulation of metabolism, redox status, signaling, and cell death pathways; the impairment of mitochondrial functions is involved in migration, invasion, and metastasis [56,57]. Similarly, the impairment of F-actin cytoskeleton assembly is crucial for the migration and invasion of metastatic cancer cells [58,59].

Actin filaments are the main structural component of the cytoskeleton and play the most important role in the structural integrity and deformability of the cell, providing the required forces for the movement and contraction of cells [60,61]. Several reports suggested that migration and invasion of cancer cells are based on the formation of actin-rich protrusions, with microfilaments together with the associated proteins mediating tumor vascularization and therefore carcinogenesis [62,63]. Ali et al., demonstrated that AuNP trapped at the nuclear membrane can stimulate the overexpression of lamin located around the nuclear membrane, thus increasing nuclear stiffness and slowing down cancer cell migration and invasion [60].

The quantitative analysis of Mitotracker Deep Red (Figure 8m) and Actin Green (Figure 8n) illustrates a trend of perturbation, respectively, for the mitochondria and the cytoskeleton actin that parallel what was found in the wound scratch assay. Specifically, the strongest change with respect to the control untreated cells was found for PdNP-treated cells, followed by the treatment with AuNP, while the treatment with AgNP, although statistically significantly different from the control in the case of mitochondria, was the less perturbative one.

## 4. Conclusions

In summary, our study on the hybrid biointerface between PVP-capped nanoparticles of gold, silver, and palladium and artificial cell membranes of SLBs or actual cells of prostate cancer (PC-3) evidenced different level of interactions for the three NP types, ruled out in terms of the different size/chemical nature of the NP but also the diverse role of PVP-capping agent around the metal core for the three investigated cases. Further systematic studies, for example, by changing the NP size and shape, as well as by scrutinizing the perturbation effect on cellular homeostasis and signaling, are needed to shed light on the precise mechanisms involved.

## Figures and Tables

**Figure 1 nanomaterials-13-01624-f001:**
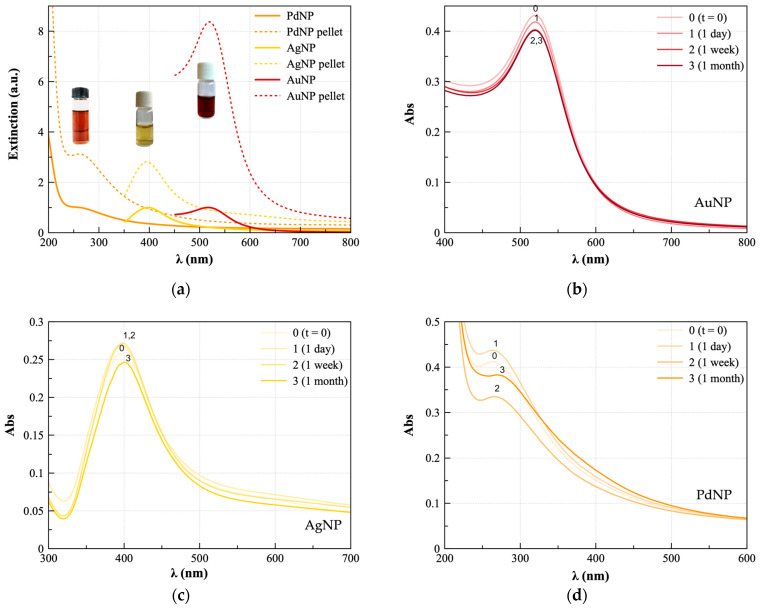
UV-vis spectra of: (**a**) as-prepared (solid lines) and purified (pellet 2, dashed lines) samples of AuNP (1× dilution), AgNP (1× dilution), and PdNP (10× dilution). Spectra of pellet 2 samples fresh (0 = immediately after the purification) and aged (1 = 1 day; 2 = 1 week; 3 = 1 month) of: (**b**) AuNP; (**c**) AgNP; (**d**) PdNP. Spectra normalized to the curve of the as-prepared sample for each NP.

**Figure 2 nanomaterials-13-01624-f002:**
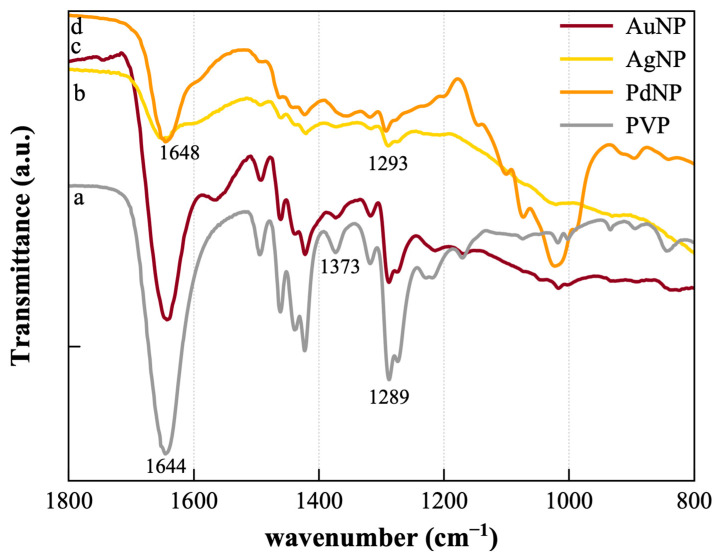
FTIR spectra of PVP (a) and PVP-capped AuNP (b), AgNP (c), and PdNP (d).

**Figure 3 nanomaterials-13-01624-f003:**
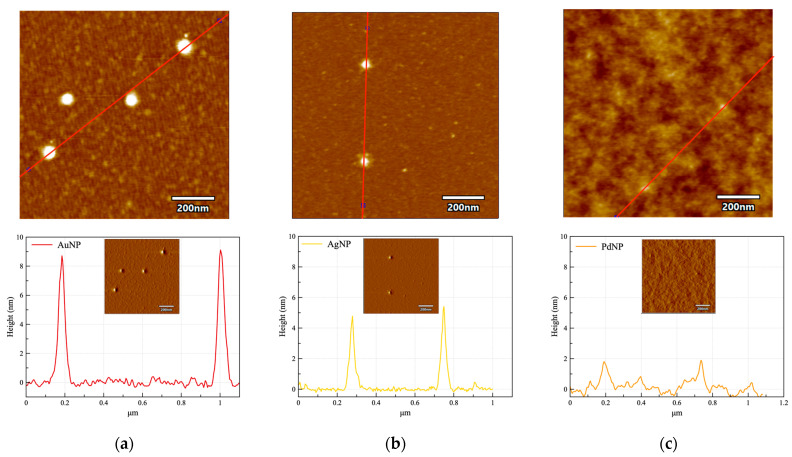
AFM representative height micrographs (top panels, z scale = 5 nm, xy scan size = 1 µm × 1 µm) with section line analysis (red line) plotted in the bottom panel graphs (insets = amplitude micrographs) for: (**a**) AuNP; (**b**) AgNP; (**c**) PdNP.

**Figure 4 nanomaterials-13-01624-f004:**
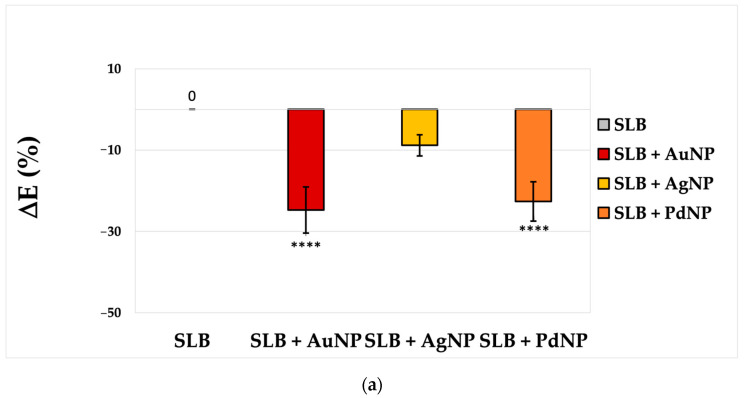
FRET analysis of NBD-Rh-labeled SLBs before and after the interaction with the three NP (50× diluted ‘pellet 2′ sample) in terms of: (**a**) percentage variation of FRET efficiency; (**b**) change of the donor-acceptor pair distance with respect to the mean value measured for the SLBs before the interaction with the NP (R = 6.4 ± 0.2 nm). (****) *p* < 0.0001, vs. bare SLB, Student’s *t*-test.

**Figure 5 nanomaterials-13-01624-f005:**
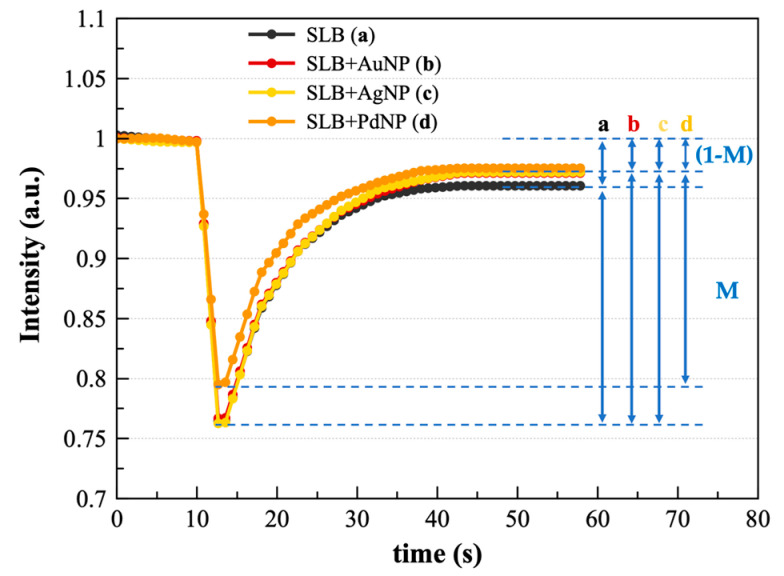
Representative recovery curves from FRAP experiments on bare SLB (a) and SBL after the interaction with AuNP (b), AgNP (c), or PdNP (d). The blue arrows point to the mobile ($M=I_{\infty }-I_{0}$) and the immobile ($1-M=I_{pre-bleach}-I_{\infty }$) fractions.

**Figure 6 nanomaterials-13-01624-f006:**
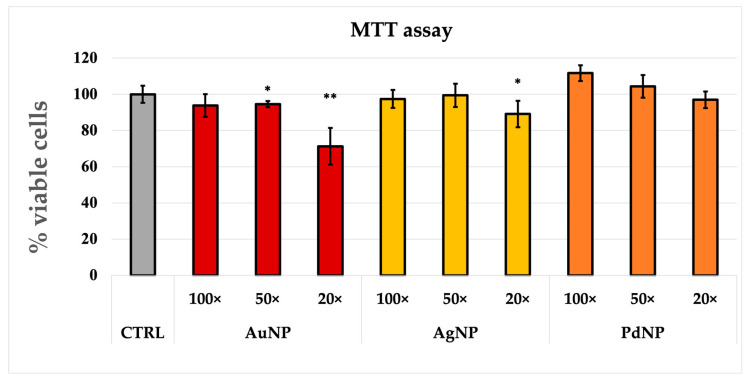
MTT assay on PC-3 cells after 24 h incubation with AuNP, AgNP, and PdNP samples. Untreated cells are also reported as the negative control. (*) *p* < 0.05, (**) *p* < 0.01 vs. untreated (CTRL), Student’s *t*-test. 20×, 50×, and 100× dilution are referred to the pellet.

**Figure 7 nanomaterials-13-01624-f007:**
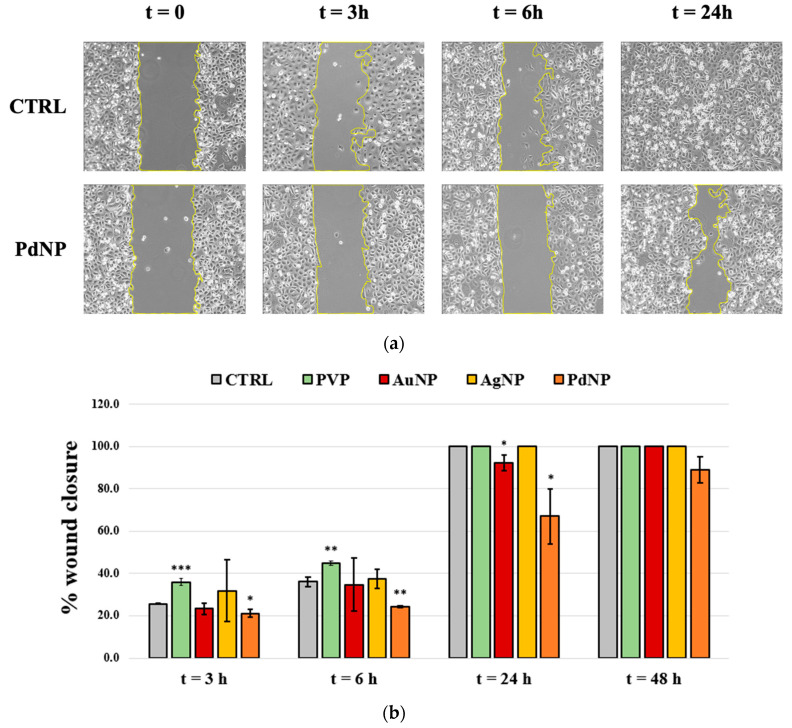
(**a**) Representative bright-field optical images of prostate cancer (PC-3) line and (**b**) quantitative analysis of cell migration after 3, 6, 24, and 48 h after the treatment with PVP, AuNP, AgNP, and PdNP samples; non-treated cells are also reported as the negative control. (*) *p* < 0.05, (**) *p* < 0.01, (***) *p* < 0.001 vs. untreated (CTRL), Student’s *t*-test.

**Figure 8 nanomaterials-13-01624-f008:**
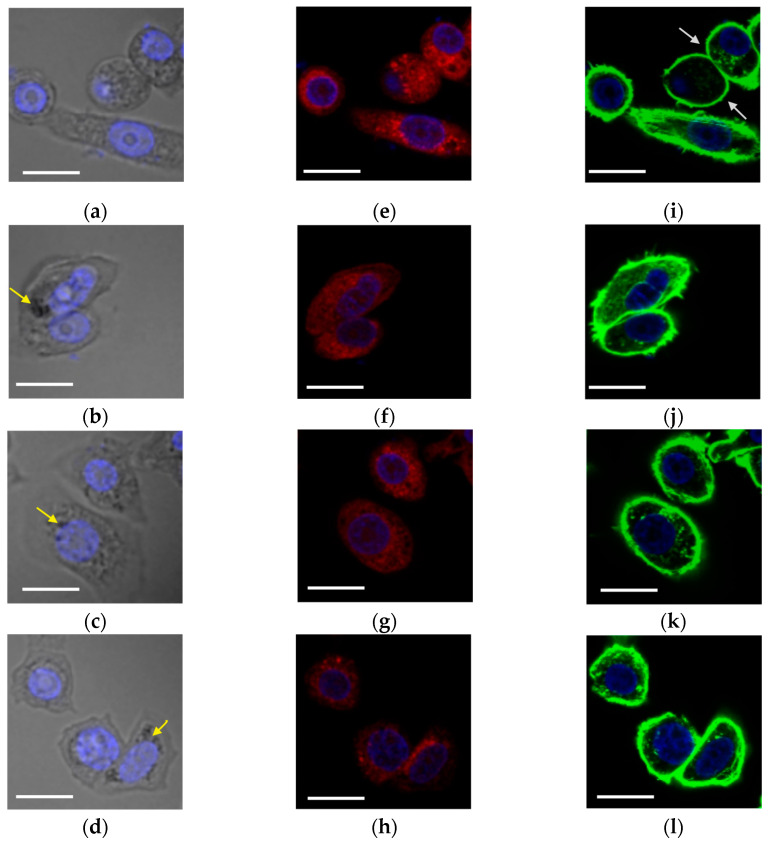
Representative LSM micrographs of PC-3 untreated (**a**,**e**,**i**) control and treated for 2 h (50× dilution of ‘pellet 2’ sample) with PVP-capped NP (**b**,**f**,**j**) AuNP; (**c**,**g**,**k**) AgNP; (**d**,**h**,**l**) PdNP). The merged micrographs of cellular DNA nuclear staining by Hoechst 33342 (blue) with optical bright field (gray, left-hand side panels; yellow arrows pointing to NP aggregates), Mitotracker Deep Red mitochondria dye (red, center panels), and Actin Green high-affinity F-actin probe (green, right-hand side panels; grey arrows pointing to thin actin filaments) are displayed. Scale bar = 20 µm. Quantitative analysis of the Mitotracker Deep Red (**m**) and Actin Green 488 (**n**) probes of mitochondria and F-actin in terms. (**) *p* < 0.01, (***) *p* < 0.001 vs. untreated (CTRL), Student’s *t*-test.

**Table 1 nanomaterials-13-01624-t001:** Calculated parameters of optical diameter (*d*_0_) and molar extinction coefficient (ε) for as- prepared (as prep.) and purified (pellet 2, p2) NP from UV-Vis analyses, from which nanoparticle concentration can be estimated according to the Lambert–Beer’s law of solutions. Measured hydrodynamic size from DLS analyses of purified NP. All values are reported as average ± standard deviation (S.D.) from three repeated measures.

Sample	*d*_0_ ± S.D. ^1^ (nm)	ε ± S.D. ^2^ (M^−1^·cm^−1^)	NP (mol·L^−1^, 10^−6^)	NP·mL^−1^
AuNP as prep.	8.1	4.9·10^7^	0.05	5.6·10^10^
AuNP p2	13.6	2.7·10^8^	0.02	2.0·10^9^
AgNP as prep.	18.7	3.1·10^9^	0.1	7.7·10^13^
AgNP p2	13.7	1.3·10^9^	3.0	1.8·10^15^
PdNP as prep.	2.2	3.4·10^5^	10.5	6.4·10^15^
PdNP p2	2.2	3.4·10^5^	12.0	7.2·10^15^

^1^ Calculated according to $d_{0}=(\lambda_{max}-515.04)/0.3647$ for AuNP [38], $d_{0}={\left[1-\left(\lambda_{max}/a\right)\right]}^{-\frac{1}{b}}$ for AgNP [39], and $d_{0}=(\lambda -253.47)/4.347$ for PdNP [29], respectively. ^2^ Calculated according to $\varepsilon_{max}=Ad^{\gamma }$ for AuNP [38], $\varepsilon =\varepsilon_{0}+Ae^{Rd}$ for AgNP [39], and to $\varepsilon =[d_{0}^{2.88}]\times \exp\left(10.48\right)$ forPdNP [40], respectively.

**Table 2 nanomaterials-13-01624-t002:** Measured values of nanoparticle size, as obtained from AFM (d_AFM_) and DLS analyses (d_H_) of purified NP (pellet 2). All values are reported as average ± standard deviation (S.D.) from three repeated measures.

Sample	d_AFM_ ± S.D. (nm)	d_H_ ± S.D. (nm)	ZP ± S.D. (mV)
AuNP	8.9 ± 0.3	29 ± 10	−0.2 ± 0.2
AgNP	5.2 ± 0.3	5 ± 2	−0.07 ± 0.2
PdNP	2.3 ± 0.1	45 ± 1	−41 ± 5

**Table 3 nanomaterials-13-01624-t003:** Analysis of the FRAP experiment for NBD-Rh-labeled SLBs before and after the interaction with the three diverse NP in terms of the calculated mobile fraction of NBD-labeled, $M_{f(NBD)}$, and rhodamine-labeled, $M_{f(Rh)}$, lipid molecules within the membrane.

Sample	*M_f_*_(*NBD*)_ ± S.D.	*M_f_*_(*Rh*)_ ± S.D.
SLB	0.89 ± 0.04	0.51 ± 0.09
SLB + AuNP	0.89 ± 0.05	0.49 ± 0.19
SLB + AgNP	0.89 ± 0.02	0.47 ± 0.16
SLB + PdNP	0.94 ± 0.20	0.72 ± 0.08

## Data Availability

Data supporting reported results are not available online.

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
