# Peer review of "Green Nanoformulations of Polyvinylpyrrolidone-Capped Metal Nanoparticles: A Study at the Hybrid Interface with Biomimetic Cell Membranes and In Vitro Cell Models"

_nanomaterials, 2023, doi:10.3390/nano13101624_

Round 1

Reviewer 1 Report

This article is primarily concerned with the synthesis of gold (Au), silver (Ag), and palladium (Pd) nanoparticles using green redox chemistry, the investigation of the physicochemical characteristics of PVP-coated nanoparticles, biophysical studies, and in vitro cellular experiments using prostate cancer (PC-3 cell line) as the study target. Although the essay is rather innovative, there are a few issues:

1. The rationale behind selecting prostate cancer as the study's focus;

2. The light color of some of the markers in Figure 1;

3. The values in Figure 2 are all between 2 and 9 nm, while the scale is 200 nm on the large side; The red lines in Figure 2 are not clearly marked;

4. The SLB in Figure 4 is 0, which is theoretically significantly different from other groups;

5. According to the paper, "recent findings suggest that PVP-based hydrogels can inhibit tumor cell migration," hence Figure 6 ought should include both a blank medium group and a blank PVP group;

6. Features shown in representative LSM micrographs in Figure 7, with arrows applied to indicate their specific variations.

Author Response

  1. The rationale behind selecting prostate cancer as the study's focus;

Answer: We thank the reviewer for this valuable comment. In the results and discussion section a paragraph, with the related references, has been introduced at the beginning of cellular analyses to justify the choice of this particular cell line.

  1. The light color of some of the markers in Figure 1;

Answer: The light colors of the curves in Figure 1 have been replaced with darker ones (and dashed lines).

  1. The values in Figure 2 are all between 2 and 9 nm, while the scale is 200 nm on the large side; The red lines in Figure 2 are not clearly marked;

Answer: As clarified in the caption (now is Figure 3), the scale bar of 200 nm refers to the scan size (xy plane), while the 0-10 nm scale values refer to the z-axis. The red lines are thicker in the revised figure.

  1. The SLB in Figure 4 is 0, which is theoretically significantly different from other groups;

Answer: As specified in the caption, the values are reported as percentage variation with respect to the control, hence the variation for control (SLB) itself is 0.

  1. According to the paper, "recent findings suggest that PVP-based hydrogels can inhibit tumor cell migration," hence Figure 6 ought should include both a blank medium group and a blank PVP group;

Answer: The figure (now 7) already contained the blank medium (negative control of untreated cells); in the revised version the results of the experiment for the treatment with PVP (positive control) have been added.

  1. Features shown in representative LSM micrographs in Figure 7, with arrows applied to indicate their specific variations.

Answer: Explicative arrows were added to the micrographs in the figure (now 8) and quoted in the text.

Reviewer 2 Report

This manuscript describes the synthesis and characterization of polyvinylpyrrolidone-capped gold, silver, and palladium nanoparticles and their biophysical and in vitro cellular interactions. The authors show that these nanoparticles have potential anti-inflammatory and anti-tumor properties, making them promising candidates for chemotherapy. The use of supported lipid bilayers and prostate cancer cell lines allows for a thorough investigation of the nanoparticles' interaction with biological systems. The manuscript provides valuable insights into the potential of green synthesized metal nanoparticles as therapeutic agents. However, some aspects of the study could be further elaborated, such as the mechanism of cellular uptake and the long-term cytotoxicity of the nanoparticles. Overall, this manuscript presents a well-executed study with promising results and could be a valuable addition to the literature in the field of nanomedicine.

Author Response

  1. However, some aspects of the study could be further elaborated, such as the mechanism of cellular uptake and the long-term cytotoxicity of the nanoparticles. Overall, this manuscript presents a well-executed study with promising results and could be a valuable addition to the literature in the field of nanomedicine.

Answer: We thank the reviewer for this valuable comment. We have added cytotoxicity studies in the revised manuscript (new Figure 6) and elaborated a discussion on the mechanisms of nanoparticle uptake in cancer cells, with some related references.

Reviewer 3 Report

Nanomaterials, # 2367150 (Alice Foti, Luana Calì, Salvatore Petralia and Cristina Satriano: Green nanoformulations of polyvinylpyrrolidone-capped metal nanoparticles: a study at the hybrid interface with biomimetic cell membranes and in vitro cell models)

The manuscript reports wide information about the preparation and cellular interactions of coated precious metal nanoparticles; however, the physical-chemical characterization of prepared nanomaterials is not complete. Consequently, the conclusion of the authors is “right” as the type, size, shape, coating influences the location in cells. With other words, they use different metals with different seed-material, size, shape(?), and coating indicating a not fully conceptualized work.

The physical chemical characterization of nanoparticles must be improved. Although, the biophysical studies of the nano-bio interfaces between PVP-capped metal NPs and SLBs do not show significant effects, these investigations indicate the advantages of FRET transfer and give inspiration for further studies between nanoparticles and highly biomimetic vesicle-like systems. I find also very positive the intercellular studies; the revelation of the intercellular location of PVP-capped metal NPs. Here one can see that the usage of fine fractions (with given type, size, coating) would be more convenient.              

In details

I suggest deducing the size of NPs with other methods too. ( XRD (wide angle Xray scattering), namely the metal seed seem to be one domains, therefore width of reflection reflects the size of a single NP), (SAXS (small angle X-ray scattering)), (EM (electron-microscopy, even in dried form on the grid), in this case the compact metal seeds and the coating can be distinguished) AFM is not a reliable method, statistically.

I suggest using a Table to show the characteristic sizes together. (DLS, AFM, data of other method)

The Figure 3 is disturbing (DLS data to other size-data, ZP does not require a separate Fig.)

I suggest infra-red spectroscopy measurements. The analyse of the vibration’s bands (CH2, especially) would give further information about the coating and compare the forms on surface of the different metals.

 In Fig 6, AuNPs and AgNPs have the same colour in the legend.

Author Response

  1. I suggest deducing the size of NPs with other methods too. (XRD (wide angle Xray scattering), namely the metal seed seem to be one domain, therefore width of reflection reflects the size of a single NP), (SAXS (small angle X-ray scattering)), (EM (electron-microscopy, even in dried form on the grid), in this case the compact metal seeds and the coating can be distinguished) AFM is not a reliable method, statistically.

Answer: We thank the reviewer for the suggestion, unfortunately, we are not able to perform the suggested analyses for the time being. However, in our manuscript, we provided the analysis of the particle size by three different methods, namely plasmonic absorption, for the optical size calculation, DLS, for the hydrodynamic size, and AFM. The results obtained with these techniques are consistent with each other.

  1. I suggest using a Table to show the characteristic sizes together. (DLS, AFM, data of other method)

Answer: we thank the referee for this suggestion; however, for the sake of clarity, we do prefer keeping the optical size, calculated from the plasmonic peaks analyses in Table 1, together with the other optical parameters. The size values estimated by AFM and DLS analyses have been included in a Table.

  1. The Figure 3 is disturbing (DLS data to other size-data, ZP does not require a separate Fig.)

Answer: Figure 3 has been removed and data moved to Table 2.

  1. I suggest infra-red spectroscopy measurements. The analyse of the vibration’s bands (CH2, especially) would give further information about the coating and compare the forms on surface of the different metals.

Answer: The FTIR analysis has been included in the revised manuscript.

  1. In Fig 6, AuNPs and AgNPs have the same colour in the legend.

Answer: In Figure 6 (now 7) the colors have been corrected.

Round 2

Reviewer 3 Report

The manuscript is improved significantly.